# Identification of Novel Therapeutic Targets for Polyglutamine Diseases That Target Mitochondrial Fragmentation

**DOI:** 10.3390/ijms222413447

**Published:** 2021-12-14

**Authors:** Annika Traa, Emily Machiela, Paige D. Rudich, Sonja K. Soo, Megan M. Senchuk, Jeremy M. Van Raamsdonk

**Affiliations:** 1Department of Neurology and Neurosurgery, McGill University, Montreal, QC H3A 2B4, Canada; annika.traa@mail.mcgill.ca (A.T.); paige.rudich@mail.mcgill.ca (P.D.R.); sonja.soo@mail.mcgill.ca (S.K.S.); 2Metabolic Disorders and Complications Program, Research Institute of the McGill University Health Centre, Montreal, QC H4A 3J1, Canada; 3Brain Repair and Integrative Neuroscience Program, Research Institute of the McGill University Health Centre, Montreal, QC H4A 3J1, Canada; 4Laboratory of Aging and Neurodegenerative Disease, Center for Neurodegenerative Science, Van Andel Research Institute, Grand Rapids, MI 49503, USA; emily.machiela@ucf.edu (E.M.); megan.senchuk@vai.org (M.M.S.); 5Division of Experimental Medicine, Department of Medicine, McGill University, Montreal, QC H4A 3J1, Canada; 6Department of Genetics, Harvard Medical School, Boston, MA 02115, USA

**Keywords:** Huntington’s disease, mitochondria, mitochondrial dynamics, polyglutamine diseases, *C. elegans*, genetics, DRP1

## Abstract

Huntington’s disease (HD) is one of at least nine polyglutamine diseases caused by a trinucleotide CAG repeat expansion, all of which lead to age-onset neurodegeneration. Mitochondrial dynamics and function are disrupted in HD and other polyglutamine diseases. While multiple studies have found beneficial effects from decreasing mitochondrial fragmentation in HD models by disrupting the mitochondrial fission protein DRP1, disrupting DRP1 can also have detrimental consequences in wild-type animals and HD models. In this work, we examine the effect of decreasing mitochondrial fragmentation in a neuronal *C. elegans* model of polyglutamine toxicity called Neur-67Q. We find that Neur-67Q worms exhibit mitochondrial fragmentation in GABAergic neurons and decreased mitochondrial function. Disruption of *drp-1* eliminates differences in mitochondrial morphology and rescues deficits in both movement and longevity in Neur-67Q worms. In testing twenty-four RNA interference (RNAi) clones that decrease mitochondrial fragmentation, we identified eleven clones—each targeting a different gene—that increase movement and extend lifespan in Neur-67Q worms. Overall, we show that decreasing mitochondrial fragmentation may be an effective approach to treating polyglutamine diseases and we identify multiple novel genetic targets that circumvent the potential negative side effects of disrupting the primary mitochondrial fission gene *drp-1*.

## 1. Introduction

Huntington’s disease (HD) is an adult-onset neurodegenerative disease caused by a trinucleotide CAG repeat expansion in the first exon of the *HTT* gene. The resulting expansion of the polyglutamine tract in the huntingtin protein causes a toxic gain of function that contributes to disease pathogenesis. HD is the most common of at least nine polyglutamine (polyQ) diseases, including spinal and bulbar muscular atrophy (SBMA), dentatorubral–pallidoluysian atrophy (DRPLA), and spinocerebellar ataxia types 1, 2, 3, 6, 7, and 17 (SCA1, SCA2, SCA3, SCA6, SCA7, and SCA17) [1,2]. Each disease occurs due to an expansion of a CAG repeat above a specific threshold number of repeats. The minimum number of disease-causing CAG repeats range from 21 CAG repeats (SCA6) to 55 CAG repeats (SCA3). These disorders are all unique neurodegenerative diseases that typically present in mid-life but can present earlier in life with larger CAG repeat expansions [3,4]. The genes responsible for these disorders appear to be unrelated, except for the presence of the CAG repeat sequence, indicating that CAG repeat expansion, independent of the genetic context, is likely sufficient to cause disease. 

Multiple lines of evidence suggest a role for mitochondrial dysfunction in the pathogenesis of polyQ diseases [5,6,7,8,9]. Both HD patients and animal models of the disease display several signs of mitochondrial dysfunction, including decreased activity in the complexes of the mitochondrial electron transport chain [10], increased lactate production in the brain [11], decreased levels of ATP production [12], lowered mitochondrial membrane potentials [13], and impaired mitochondrial trafficking [14]. While less well studied than HD, other polyQ diseases also have evidence of mitochondrial deficits [15,16,17,18]. 

Mitochondrial fragmentation is a consistent feature of HD as it occurs in HD cell lines, HD worm models, HD mouse models, and cells derived from HD patients [19,20,21,22,23,24,25,26,27]. Mitochondrial fragmentation has also been observed in models of other polyQ diseases, including SCA3, SCA7, and SBMA [28,29,30]. This suggests that CAG repeat expansion may be sufficient to cause mitochondrial fragmentation.

In order to decrease HD-associated mitochondrial fragmentation, multiple groups have targeted the mitochondrial fission protein DRP1 in models of HD. While genetic or pharmacologic treatments that either directly or indirectly inhibit DRP1 activity typically exhibit beneficial effects in HD models [19,20,22,31,32,33], the disruption of DRP1 has also been found to exacerbate disease phenotypes [26]. The difference in effect may be due to the level of DRP1 disruption, as deletion of *drp-1* was detrimental in an HD model, while RNAi knockdown of *drp-1* in the same model had mixed effects [26]. Decreasing DRP1 levels can also be detrimental in a wild-type background [26,34,35,36,37,38,39]. Thus, reducing mitochondrial fragmentation through other genetic targets may be a more ideal therapeutic strategy for HD and other polyQ diseases than disrupting DRP1.

In this work, we show that CAG repeat expansion is sufficient to disrupt mitochondrial morphology and function in a neuronal model of polyQ toxicity. The neuronal model of polyQ toxicity also displays deficits in movement and lifespan, which are ameliorated by the deletion of *drp-1*. Using this model, we performed a targeted RNAi screen and identified eleven novel genetic targets that improve movement and increase lifespan. Overall, this work demonstrates that decreasing mitochondrial fragmentation may be an effective therapeutic strategy for polyQ diseases and identifies multiple potential genetic therapeutic targets for these disorders.

## 2. Results

### 2.1. Mitochondrial Morphology Is Disrupted in a Neuronal Model of Polyglutamine Toxicity

In order to study the effect of polyQ toxicity on mitochondrial dynamics in neurons, we utilized a model that expresses a polyQ protein containing 67 glutamines tagged with YFP under the pan-neuronal *rgef-1* promoter [40]. Henceforth, these worms will be referred to as Neur-67Q worms. While this model has been studied previously, mitochondrial morphology and function in these worms have not been characterized. To visualize mitochondrial morphology in GABAergic neurons, we generated a new strain expressing mScarlet fused with the N-terminus of TOMM-20 (translocase of outer mitochondrial membrane 20), thus targeting the red fluorescent protein mScarlet to the mitochondria. In the rest of the paper, these worms (*rab-3p::tomm-20::mScarlet*) are referred to as mito-mScarlet worms. After crossing Neur-67Q worms to mito-mScarlet worms, we examined mitochondrial morphology in the dorsal nerve cord in day 1 adult worms.

We found that day 1 adult Neur-67Q worms exhibit mitochondrial fragmentation (Figure 1A). Compared to mito-mScarlet control worms, Neur-67Q; mito-mScarlet worms have a decreased number of mitochondria (Figure 1B). Although the mitochondrial area is not significantly affected by CAG repeat expansion at day 1 of adulthood (Figure 1C), Neur-67Q worms exhibit a decreased axonal mitochondrial load (Figure 1D), which is calculated as mitochondria area per length of the axon. In addition, the shape of the mitochondria is affected as Neur-67Q; mito-mScarlet worms have more circular mitochondria (Figure 1E) and a decreased maximum Feret’s diameter of the mitochondria (Figure 1F), which is the maximum distance between two parallel tangents to the mitochondria.

### 2.2. Differences in Mitochondrial Morphology in Neuronal Model of Polyglutamine Toxicity Are Exacerbated with Increasing Age

To determine the effect of age on mitochondrial dynamics in Neur-67Q worms, we imaged and quantified mitochondrial morphology in worms at day 7 of adulthood. As in young adult worms, adult day 7 Neur-67Q worms exhibit mitochondrial fragmentation and a decrease in axonal mitochondria, which is much greater than observed in day 1 adult worms (Figure 1A). Quantification of mitochondrial morphology revealed that day 7 Neur-67Q worms have a decreased mitochondrial number (Figure 1G), decreased mitochondrial area (Figure 1H), decreased axonal mitochondrial load (Figure 1I), increased mitochondrial circularity (Figure 1J), and decreased Feret’s diameter of the mitochondria (Figure 1K). These results indicate that aged Neur-67Q worms have a highly disconnected mitochondrial network morphology. Furthermore, the percentage decreases in mitochondrial number (−27% day 1 versus −64% day 7), mitochondrial area (−10% day 1 versus −22% day 7), and axonal mitochondrial load (−35% day 1 versus −70% day 7) are all much greater at day 7 than at day 1, indicating that the deficits in mitochondrial morphology in Neur-67Q worms worsen with age (Appendix A, see Appendix A).

### 2.3. Neuronal Model of Polyglutamine Toxicity Exhibits Altered Mitochondrial Function

To determine if the differences in mitochondrial morphology that we observed affect mitochondrial function, we measured the rate of oxidative phosphorylation (oxygen consumption) and energy production (ATP levels) in day 1 adult worms. We found that Neur-67Q worms have increased oxygen consumption (Figure 2A) but decreased levels of ATP (Figure 2B). This suggests that the mitochondria in Neur-67Q are less efficient than in wild-type worms, possibly due to mitochondrial uncoupling. Combined, these results show that the presence of a disease-length CAG repeat expansion is sufficient to disrupt mitochondrial morphology and function.

### 2.4. Disruption of Mitochondrial Fission Is Beneficial in a Neuronal Model of Polyglutamine Toxicity

As disruption of *drp-1* has been shown to ameliorate phenotypic deficits in various models of HD, we examined whether disruption of *drp-1* would be beneficial in Neur-67Q worms. We found that deletion of *drp-1* significantly improved mobility (Figure 3A) and increased lifespan (Figure 3B) in Neur-67Q worms. While the *drp-1* deletion decreased fertility (Figure 3C) and slowed development (Figure 3D) in wild-type worms, it did not affect either of these phenotypes in Neur-67Q worms. This may be because mitochondrial morphology is already different than wild-type in Neur-67Q worms, while disruption of *drp-1* in wild-type worms results in hyperfused mitochondria.

Finally, we examined the effect of *drp-1* deletion on mitochondrial function in Neur-67Q worms. We found that the increased oxygen consumption observed in Neur-67Q worms is significantly decreased by disruption of *drp-1* (Figure 3E). However, the *drp-1* deletion was unable to increase the low ATP levels in Neur-67Q worms and decreased ATP levels in wild-type worms (Figure 3F). Although the effects of *drp-1* deletion in Neur-67Q worms are primarily beneficial, the loss of *drp-1* increased expression of the disease-length polyQ mRNA (Appendix A), as we and others have previously observed [19,26], which would be predicted to cause increased toxicity. 

To ensure that the beneficial effects of the *drp-1* deletion in Neur-67Q worms are caused by the disruption of *drp-1*, we examined the effect of *drp-1* RNAi in Neur-67Q worms. Because most *C. elegans* neurons are resistant to RNAi knockdown [41], we first crossed Neur-67Q worms to a worm strain that exhibits enhanced RNAi knockdown specifically in the neurons but is resistant to RNAi in other tissues [42]. In the resulting strain (Neur-67Q;*sid-1;unc-119p::sid-1*), RNAi is only active in the nervous system.

As with the *drp-1* deletion, knocking down *drp-1* expression throughout life increased the rate of movement (Appendix A) and increased lifespan (Appendix A) in Neur-67Q worms, while having no effect on fertility in Neur-67Q worms (Appendix A). As with the *drp-1* deletion, *drp-1* RNAi decreased both oxygen consumption and ATP levels in Neur-67Q worms (Appendix A).

### 2.5. Disruption of Mitochondrial Fission Decreases Mitochondrial Fragmentation in Neurons

Having shown that *drp-1* deletion ameliorates phenotypic deficits in Neur-67Q worms, we wondered whether the alterations in mitochondrial morphology were also corrected. Accordingly, we imaged and quantified mitochondrial morphology in Neur-67Q; *drp-1* worms at day 1 (Appendix A) and day 7 (Figure 4) of adulthood. On day 1 of adulthood, disruption of *drp-1* markedly elongated the neuronal mitochondria, leading to decreased mitochondrial fragmentation in both Neur-67Q worms and wild-type worms (Appendix A). Quantification of these differences revealed that deletion of *drp-1* results in significantly decreased numbers of mitochondria (Appendix A), significantly increased mitochondrial area (Appendix A), significantly increased axonal mitochondrial load (Appendix A), significantly decreased mitochondrial circularity (Appendix A), and significantly increased Feret’s diameter (Appendix A).

The beneficial effects of *drp-1* disruption on mitochondrial morphology in Neur-67Q worms are also observed at day 7 of adulthood (Figure 4A). In day 7 adult Neur-67Q worms, disruption of *drp-1* increases mitochondrial number (Figure 4B), mitochondrial area (Figure 4C), and axonal mitochondrial load (Figure 4D), while decreasing mitochondrial circularity (Figure 4E) and increasing the Feret’s diameter of the mitochondria (Figure 4F). Similar changes are observed in wild-type worms with the exception of mitochondrial number, which is significantly decreased by *drp-1* disruption (Figure 4B). 

Combined, these results indicate that *drp-1* has a beneficial effect on mitochondrial morphology in Neur-67Q worms. Interestingly, CAG repeat expansion in Neur-67Q worms has no effect on mitochondrial morphology in the *drp-1* mutant background (Appendix A).

### 2.6. Targeting Genes That Affect Mitochondrial Fragmentation Improves Thrashing Rate and Lifespan in a Neuronal Model of Polyglutamine Toxicity

While our results show that decreasing levels of *drp-1* are beneficial in a neuronal worm model of polyQ toxicity, this treatment had a detrimental effect in a *C. elegans* model of HD in which exon 1 of mutant huntingtin is expressed in the body wall muscle [26]. Moreover, a number of studies have found that disruption of DRP1 can be detrimental in organisms ranging from worms to humans [34,35,36,37,38,39].

To circumvent potential detrimental effects of disrupting *drp-1*, we targeted other genes that have been previously found to decrease mitochondrial fragmentation [43]. In the previous study, a targeted RNAi screen identified 24 mitochondria-related RNAi clones that decrease mitochondrial fragmentation in the body wall muscle of *C. elegans*. We examined the effect of these 24 RNAi clones in neuron-specific RNAi Neur-67Q worms (Neur-67Q; *sid-1; unc-119p::sid-1*). Treatment with RNAi was begun at the L4 stage of the parental generation, and the rate of movement was assessed in the progeny (experimental generation). 

We found that 16 of the 24 RNAi clones that decrease mitochondrial fragmentation significantly increased the rate of movement in the neuron-specific RNAi Neur-67Q model (Figure 5A). To ensure that the improved movement in Neur-67Q worms did not result from a general effect of these RNAi clones on the rate of movement, we treated *sid-1;unc-119p::sid-1* control worms with the same panel of 24 RNAi clones and examined movement. Unlike the Neur-67Q worms, we found that only four of the RNAi clones improved movement in the control neuron-specific RNAi strain (Figure 5B). This indicates that, for the majority of the RNAi clones that show a benefit, the improvement in movement is specific to the neuronal model of polyQ toxicity. 

We next examined whether the genes that improved motility in neuron-specific RNAi Neur-67Q worms also improved longevity. We found that 11 of the 16 RNAi clones that increased the rate of movement also increased lifespan in neuron-specific RNAi Neur-67Q worms (Figure 6). In contrast, only three of these RNAi clones increased lifespan in the neuron-specific RNAi control strain (Appendix A). Overall, RNAi clones that decrease mitochondrial fragmentation in body wall muscle are beneficial in a neuronal model of polyQ toxicity. 

## 3. Discussion

Since the discovery of the genes responsible for HD and other polyQ diseases [44,45], multiple animal models of these disorders have been generated to gain insight into disease pathogenesis [46,47]. This includes *C. elegans* models of HD and polyQ toxicity [40,48,49,50]. *C. elegans* offers a number of advantages for studying neurodegenerative diseases, including being able to perform large-scale screens for disease modifiers rapidly and cost-effectively [51,52]. In addition, the interconnections of all of the neurons in *C. elegans* have been mapped. In terms of studying mitochondrial dynamics, the transparent nature of *C. elegans* facilitates imaging mitochondrial morphology in a live organism, which can then be correlated with whole-organism phenotypes.

### 3.1. CAG Repeat Expansion Disrupts Mitochondrial Morphology and Function in Neurons

HD and other polyQ diseases are neurodegenerative diseases in which the most severe pathology occurs in neurons. We previously examined mitochondrial fragmentation in a muscle model of HD as it is more experimentally accessible [26]. However, to gain greater physiological relevance, in this study, we generated novel strains to examine mitochondrial morphology in neurons. We found that CAG repeat expansion in Neur-67Q worms is sufficient to cause mitochondrial fragmentation neurons, as well as a progressive decrease in the abundance of mitochondria in the axons of the dorsal nerve cord. The differences in mitochondrial number, axonal load, size, circularity, and length (Feret’s diameter) in the neuronal model of polyQ toxicity are quantifiable and highly significant. 

In these experiments, we used wild-type worms as a control rather than worms expressing a CAG repeat tract within the unaffected range. As a result, we can’t exclude the possibility that mitochondrial fragmentation might also be caused by the expression of short CAG repeat sequences. However, we think that this is unlikely as we have previously found that expression of CAG repeat expansions of 24–28 does not lead to mitochondrial fragmentation in the body wall muscle [26]. Similarly, others have only observed mitochondrial fragmentation with disease length CAG repeat sequences [22]. 

Importantly, Neur-67Q worms also exhibited changes in mitochondrial function, including a significant increase in oxygen consumption and a significant decrease in ATP levels. These differences are particularly striking given that oxygen consumption and ATP levels were measured in whole worms while the expanded polyQ transgene is only expressed in neurons, which make up 302 of the worm’s 959 cells. Given the magnitude of the differences observed, it is possible that changes occurring in the neurons are having cell-non-autonomous effects on mitochondrial function in other tissues. 

Although the yield of ATP from oxidative phosphorylation is variable [53], oxygen consumption and ATP production normally correlate under basal conditions due to the high dependence of ATP production on the electron transport chain in *C. elegans* [54,55]. The opposing changes in ATP and oxygen consumption suggest that the mitochondria in Neur-67Q worms are inefficient or damaged, leading to a marked decrease in the ATP produced per amount of oxygen consumed. We observed a similar pattern in a mitophagy-defective worm model of Parkinson’s disease in which there is a deletion of *pdr-1/PRKN* [56].

It is interesting to note that *drp-1* deletion was still able to improve movement in Neur-67Q worms despite further decreasing the levels of ATP. Similarly, *drp-1* deletion decreased ATP levels in wild-type worms but did not decrease their movement. These findings suggest that the movement deficit in Neur-67Q worms is not simply a result of decreased levels of ATP but more likely due to a disruption of neuronal function. 

### 3.2. Tissue-Specific Effects of Disrupting Mitochondrial Fission

One of the most surprising findings of our current study is that deletion of *drp-1* has different effects in neuronal and body wall muscle models of polyQ toxicity (see Appendix A for comparison). In the neuronal model, deletion of *drp-1* increases movement and lifespan and has no detrimental effect on development or fertility. In contrast, disruption of *drp-1* in the body wall muscle model decreases movement, lifespan, fertility, and the rate of development [26]. The opposing effects of reducing *drp-1* on polyQ toxicity in neurons compared to body wall muscle suggest that the optimal balance between mitochondrial fission and fusion may differ between tissues. Alternatively, it is possible that the loss of mitochondrial fission is better tolerated in neurons than in body wall muscle, even though both tissues are post-mitotic in *C. elegans*. Finally, it could be that decreasing *drp-1* levels is beneficial in neurons because it is more effective at correcting disruptions in mitochondrial networks in that tissue (Figure 4) than in body wall muscle, where *drp-1* deletion had little or no effect on mitochondrial morphology [26]. 

It should be noted that the neuronal model of polyQ toxicity used in this study and the HD muscle model that we utilized previously cannot be directly compared due to differences between these strains beyond the tissue of expression (Appendix A). Notably, BW-Htt-74Q worms have a small fragment of the huntingtin protein linked to the expanded polyQ tract, while Neur-67Q only have the expanded polyQ tract. The size of the polyQ tract is different between these two strains, and BW-Htt-74Q worms have the polyQ tagged with GFP, while the polyQ is tagged with YFP in Neur-67Q worms. Thus, while our results do not rule out other factors contributing to the differences between the neuronal strain and the muscle strain, they clearly show that decreasing *drp-1* levels can be beneficial in worms expressing an expanded polyQ tract in neurons, and that decreasing *drp-1* levels can be detrimental in worms expressing an expanded polyQ tract in muscle cells. 

### 3.3. Decreasing Mitochondrial Fragmentation as a Therapeutic Strategy for Polyglutamine Diseases

Due to the many roles *drp-1* plays in promoting proper cellular function through control of the mitochondria and the previously observed detrimental effects of decreasing *drp-1* in a body wall muscle model [26], decreasing levels or activity of DRP-1 may be a non-ideal therapeutic target for HD or other polyQ diseases. Accordingly, we explored other possible genetic targets that decrease mitochondrial fragmentation. We performed a targeted RNAi screen using 24 RNAi clones previously found to decrease mitochondrial fragmentation in the body wall muscle [43]. A high percentage of these RNAi clones increased movement (16 of 24 RNAi clones that decrease fragmentation) and lifespan (11 of 16 RNAi clones that improve movement) in Neur-67Q worms. 

As we obtained numerous positive hits, we did not confirm knockdown by qPCR or confirm a decrease in mitochondrial fragmentation. Thus, we can’t exclude the possibility that the remaining eight genes that failed to show a beneficial effect may have had either insufficient genetic knockdown or did not exhibit the predicted effect on mitochondrial morphology. Nonetheless, a high proportion of RNAi clones previously found to decrease mitochondrial fragmentation increased movement in the neuronal HD model indicating that multiple genetic approaches to decreasing mitochondrial fragmentation are beneficial in worm models of polyQ toxicity.

In order to prioritize therapeutic targets for further characterization and validation, we analyzed the results from the current study with our previous study of these RNAi clones in a body wall muscle model of HD [26] (Table 1). The genes were ranked by giving one point for improving either: thrashing rate in Neur-67Q worms; lifespan in Neur-67Q worms; the crawling rate in BW-Htt74Q worms; or thrashing rate in BW-Htt74Q worms. Of the 24 RNAi clones tested in both models, 21 clones exhibited a beneficial effect on at least one phenotype. This indicates that multiple approaches to decreasing mitochondrial fragmentation can ameliorate deficits caused by CAG repeat expansion. The top-ranked therapeutic targets were *alh-12* and *pgp-3*, which resulted in improvement of all four assessments, and *gpd-4, immt-2, sdha-2* and *wht-1*, which resulted in improvement in three of the assessments (Table 1). As the RNAi clones targeting *alh-12* and *pgp-3* were the top-ranked hits, we confirmed that these RNAi clones successfully knocked down the expression of *alh-12* and *pgp-3*, respectively (Figure 7). Interestingly, knockdown of *alh-12* or *pgp-3* has detrimental effects on lifespan or movement in control strains, indicating that their beneficial effect is specific to animals with disease-length CAG repeats. 

The *alh-12* gene encodes a cytoplasmic aldehyde dehydrogenase that is expressed in the intestine, body wall muscle, and specific neurons. It is involved in multiple metabolic pathways, including arginine metabolism, glycerolipid metabolism, glycolysis/gluconeogenesis, and tryptophan degradation. As very little is known about the functions of ALH-12, it is hard to speculate how disrupting *alh-12* may be acting to improve movement and lifespan in the worm models of polyQ toxicity. The human homolog of ALH-12, ALDH9A1 can be inhibited by diethylaminobenzaldehyde [57], which can potentially be used to validate the neuroprotective effects of ALH-12 inhibition in mammalian models.

The *pgp-3* gene encodes a p-glycoprotein related protein. It is a transmembrane protein that transports molecules out of the cytoplasm. PGP-3 is primarily expressed in the intestine [58], but has also been reported in other tissues. Disruption of *pgp-3* sensitizes worms to *P. aeruginosa* in a toxin-based fast kill assay [59], as well as exposure to colchicine and chloroquinone [60], presumably by disrupting the active removal of the toxic compounds from cells. While it seems counterintuitive that loss of a protective function against toxins and xenobiotics is protective against polyQ toxicity, knockdown of *pgp-3* may be acting through hormesis, the process by which exposure to mild stress activates protective pathways that can increase resistance to subsequent stresses and extend longevity. Exposing worms to mild stress (e.g., heat stress) increases both stress resistance and lifespan [61]. Thus, it is possible that preventing the transport of specific molecules out of the cytoplasm through disruption of *pgp-3* induces mild stress, which leads to activation of protective stress response pathways. Alternatively, it could be that retention of specific molecules in the cytoplasm somehow protects against the toxic effects of the CAG repeat expansion. 

As relatively little is known about *alh-12* and *pgp-3*, it will be important to further characterize their biological functions in order to gain insight into mechanisms of neuroprotection. As both genes have homologs in mice and humans, it will also be important to validate these genes in mammalian models to determine if their ability to protect against polyQ disease is conserved across species. Demonstrating a protective effect in mammalian models would provide strong support for these genes as potential therapeutic targets in polyQ diseases.

## 4. Materials and Methods

### 4.1. Strains

N2 (WT)AM102 *rmIs111[rgef-1p::40Q:YFP]* referred to as Neur-40QAM717 *rmIs284[rgef-1p::67Q:YFP]* referred to as Neur-67QJVR258 *drp-1*(*tm1108*);*rmIs284[rgef-1p::67Q:YFP]*JVRV438 *rmIs284[rgef-1p::67Q:YFP]; sid-1(pk3321); uIs69 [pCFJ90 (myo-2p::mCherry) + unc-119p::sid-1]*JVR443 *rmIs284[rgef-1p::67Q:YFP]; uIs69 [pCFJ90 (myo-2p::mCherry) + unc-119p::sid-1]*PHX3820 *sybIs3820[rab-3p::tomm-20::mScarlet]* referred to as mito-mScarletJVR611 *rmIs284[rgef-1p::67Q:YFP];drp-1(tm1108); sybIs3820[rab-3p::tomm-20::mScarlet]* referred to as Neur-67Q;*drp-1;*mito-mScarletJVR612 *rmIs284[rgef-1p::67Q:YFP]; sybIs3820[rab-3p::tomm-20::mScarlet]* referred to as Neur-67Q;mito-mScarletJVRV613 *drp-1(tm1108); sybIs3820[rab-3p::tomm-20::mScarlet]* referred to as *drp-1;*mito-mScarletMQ17V53 *drp-1* (*tm1108*)TU3401 *sid-1(pk3321); uIs69 [pCFJ90 (myo-2p::mCherry) + unc-119p::sid-1]*

Strains were maintained at 20 °C on NGM plates seeded with OP50 bacteria. The Neur-67Q model of HD is an integrated line that has been well characterized previously [40]. All crosses were confirmed by genotyping using PCR for deletion mutations, sequencing for point mutations, and confirmation of fluorescence for fluorescent transgenes.

### 4.2. Generation of Strains to Monitor Mitochondrial Morphology in GABA Neurons

The *rab-3p::tomm-20::mScarlet* strain was generated by SunyBiotech Co., Ltd., Fujian, China, The 1208 bp *rab-3* promoter sequence (Addgene Plasmid #110880) was inserted directly upstream of the N-terminal TOMM-20 coding region. The first 47 amino acids of TOMM-20 were connected through a flexible linker (3xGGGGS) to the N-terminal of wrmScarlet [62]. The strain was generated through microinjection of *rab-3p::tomm-20::mScarlet* in the pS1190 plasmid (20 ng/µL) into wild-type N2 worms. The transgenic strain was integrated by γ-irradiation and the outcrossed 5X to remove background mutations.

### 4.3. Confocal Imaging and Quantification

Mitochondrial morphology was imaged and quantified using worms that express mitochondrially-targeted mScarlet specifically in neurons (*rab-3p::tomm-20::mScarlet*). Worms at day 1 or day 7 of adulthood were mounted on 2% agar pads and immobilized using 10 µM levamisole. Worms were imaged under a 63× objective lens on a Zeiss LSM 780 confocal microscope. All conditions were kept the same for all images. Single plane images were collected for a total of twenty-five young adult worms over three biological replicates for each strain. Quantification of mitochondrial morphology was performed using ImageJ. Segmentation analysis was carried out using the SQUASSH (segmentation and quantification of subcellular shapes) plugin. Particle analysis was then used to quantify number of mitochondria, mitochondrial area, axonal mitochondrial load, mitochondrial circularity, and maximum Feret’s diameter (an indicator of particle length). Axonal load was calculated as the total mitochondrial area (μm^2^) in a region of interest (ROI), per length (μm) of axon in the ROI. For representative images, mScarlet and YFP channels were merged. We observed some bleed-through of YFP into the red channel for strains expressing the 67Q-YFP transgene. Particles that showed up in the mScarlet images as a result of YFP bleed-through were manually excluded from morphology quantification based on the numbered particle mask output from the ImageJ particle analyzer.

### 4.4. Oxygen Consumption

To measure basal oxygen consumption, a Seahorse XF_e_96 analyzer (Seahorse bioscience Inc., North Billerica, MA, USA) [57] was used. Adult day 1 worms were washed in M9 buffer (22 mM KH_2_PO_4_, 34 mM NA_2_HPO_4_, 86 mM NaCl, 1 mM MgSO_4_) and pipetted in calibrant (~50 worms per well) into a Seahorse 96-well plate. Oxygen consumption rate was measured six times. One day before the assay, well probes were hydrated in 175 µL of Seahorse calibrant solution overnight. The heating incubator was turned off to allow the Seahorse machine to reach room temperature before placing worms inside. Rates of respiration were normalized to the number of worms per well. Plate readings were within 20 min of introducing the worms into the well and normalized relative to the number of worms per well.

### 4.5. ATP Production

To measure ATP production, a luminescence-based ATP kit was used [63]. Approximately 200 age-synchronized worms were collected in deionized water before being washed and freeze-thawed three times. A Bioruptor (Diagenode) was used to sonicate the worm pellet for 30 cycles of alternating 30 s on and 30 s off. The pellet was boiled for 15 min to release ATP and then centrifuged at 11,000× *g* for 10 min at 4 °C before the resulting supernatant was collected. A Molecular Probes ATP determination Kit (Life Technologies) was used to measure ATP. Luminescence was normalized to protein levels determined by a Pierce BCA protein determination kit (Thermo Scientific, Waltham, MA, USA).

### 4.6. Rate of Movement

To measure rate of movement, thrashing rate in liquid was assessed using video-tracking and computer analysis [64]. Approximately 50 day 1 adult worms were placed in M9 buffer on an unseeded NGM plate. An Allied Vision Tech Stingray F-145 B Firewire Camera (Allied Vision, Exton, PA, USA) was used to capture videos at 1024 × 768 resolution and 8-bit using the MATLAB image acquisition toolbox. The wrMTrck plugin for ImageJ (http://www.phage.dk/plugins) was used to analyze rate of movement.

### 4.7. Lifespan

To measure lifespan, worms were placed on nematode growth media (NGM) agar plates containing 25 μM 5-fluoro-2′-deoxyuridine (FUdR). FUdR was used to reduce progeny development. At a 25 μM concentration of FUdR, progeny development into adulthood is not completely prevented in the first generation so animals were transferred to fresh plates after 4 days [65]. Worms were moved to fresh plates weekly and survival was observed by gentle prodding every 2 days. Lifespan experiments were conducted with three replicates of 30 worms each.

### 4.8. Brood Size

To determine brood size, individual young adult worms were placed onto agar plates and transferred every day to new plates until progeny production stopped. Plates of resulting progeny were quantified when adulthood was reached. Experiments were conducted with three replicates of five worms each. 

### 4.9. Post-Embryonic Development

To measure post-embryonic development (PED), eggs were moved to agar plates and left to hatch for 3 h. L1 worms that were newly hatched were transferred to a new plate. The PED time was considered the total time from hatching to the young adult stage. Experiments were conducted with three replicates of 20 animals each. 

### 4.10. Quantitative Reverse-Transcription PCR (qPCR)

To quantify mRNA levels, pre-fertile young adult worms were harvested in Trizol as previously described [66]. Three biological replicates for N2, BW-40Q, and BW-Htt74Q worms were collected to quantify gene expression. A High-Capacity cDNA Reverse Transcription kit (Life Technologies/Invitrogen) was used to convert mRNA to cDNA. A FastStart Universal SYBR Green kit (Roche) in an AP Biosystems real-time PCR machine was used to perform qPCR [67,68]. Primer sequences used:
*yfp* (L-GACGACGGCAACTACAAGAC, R-TCCTTGAAGTCGATGCCCTT);*pgp-3* (L-CTGTCTGGTGGACAGAAGCA, R-AAGAGCTGACGTGGCTTCAT);*alh-12* (L-GCCTTCAAGCTGGAACTGTTT, R-TTGCCTTTGTCTGAGTATGAGC).

### 4.11. RNAi

To knockdown gene expression, sequence-verified RNAi clones from the Ahringer RNAi library were grown approximately 12 h in LB with 50 μg/mL carbenicillin. Bacteria cultures were 5× concentrated and seeded onto NGM plates containing 5 mM IPTG and 50 μg/mL carbenicillin. Plates were incubated at room temperature for 2 days to induce RNAi. For the L4 parental paradigm, in which RNAi knockdown began in the parental generation, L4 worms were plated on RNAi plates for one day and then transferred to a new plate as gravid adults. After 24 h, the worms were removed from the plates. The resulting progeny from these worms were analyzed. RNAi experiments were conducted at 20 °C. 

### 4.12. Experimental Design and Statistical Analysis

All experiments were performed with experimenters blinded to the genotype of the worms. Worms used for experiments were randomly selected from maintenance plates. A minimum of three biological replicates, in which independent populations of worms tested on different days, were performed for each experiment. Automated computer analysis was performed in assays where possible to eliminate potential bias. Power calculations were not used to determine sample size for experiments, since the sample size used in *C. elegans* studies are typically larger than required for observing a difference that is statistically significant. Three biological replicates were used for measurements of mitochondrial morphology. At least six replicates of ~50 worms each were used for measurements of oxygen consumption. Three biological replicates of ~200 worms each were used for ATP measurements. Three biological replicates of a 60 mm plate of worms were used for mRNA measurements. At least three biological replicates of ~40 worms each were used for the thrashing assays. Three biological replicates of thirty worms each were used for lifespan assays. Six individual worms were used for measuring brood size. Three biological replicates of twenty-five worms each were used to measure post-embryonic development time. GraphPad Prism was used to perform statistical analysis. One-way, two-way, or repeated-measures ANOVA were used to determine statistically significant differences between groups with Dunnett’s or Bonferroni’s multiple comparisons test. For analysis of lifespan, Kaplan-Meier survival plots were graphed, and the Log-rank test was used to determine significant differences between the two groups. This study has no pre-specified primary endpoint. Sample size calculations were not performed.

## 5. Conclusions

In this study, we showed that a *C. elegans* neuronal model of polyQ toxicity exhibits deficits in mitochondrial morphology and function, which are associated with decreased movement and lifespan. Decreasing the levels of the mitochondrial fission gene *drp-1* through genetic deletion or RNAi increases both movement and lifespan in Neur-67Q worms. Similarly, treatment of Neur-67Q worms with RNAi clones that decrease mitochondrial fragmentation results in increased movement and lifespan. Overall, this work suggests that decreasing mitochondrial fragmentation may be beneficial in treating HD and other polyQ diseases and identifies alternative genetic targets that circumvent the negative effects of disrupting DRP-1. Future studies will be needed to further investigate the mechanisms by which the genes we identified are beneficial and to validate these targets in other models of polyQ diseases.

## Figures and Tables

**Figure 1 ijms-22-13447-f001:**
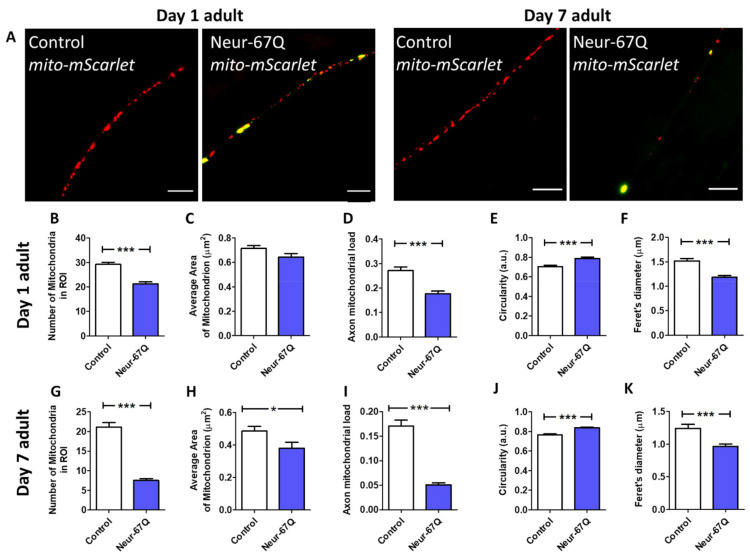
CAG repeat expansion disrupts mitochondrial morphology in neurons. Representative images of mitochondria (red) in the dorsal nerve cord in control and Neur-67Q worms at day 1 and day 7 of adulthood demonstrate that Neur-67Q worms exhibit mitochondrial fragmentation and decreased numbers of mitochondria in neurons (**A**). Mitochondrial morphology was visualized by fusing the red fluorescent protein mScarlet to the mitochondrially-targeted protein TOMM-20 in order to target mScarlet to the mitochondria. Yellow fluorescence is bleed-through from the 67Q::YFP protein. Scale bar indicates 10 µM. Quantification of mitochondrial morphology at day 1 of adulthood reveals that Neur-67Q worms have a decreased number of mitochondria compared to control worms (**B**). While mitochondrial area is not significantly affected in Neur-67Q worms at day 1 of adulthood (**C**), these worms have a significant decrease in axonal mitochondrial load (**D**) compared to control worms. The mitochondria of day 1 adult Neur-67Q worms have increased circularity (**E**) and a decreased Feret’s diameter (**F**) compared to the mitochondria of control worms. Similarly, Neur-67Q worms at day 7 of adulthood have a significantly decreased number of mitochondria compared to control worms (**G**). Day 7 adult Neur-67Q worms also exhibit decreased mitochondrial area (**H**), decreased axonal mitochondrial load (**I**), increased mitochondrial circularity (**J**), and a decreased mitochondrial Feret’s diameter (**K**) compared to control worms. Control worms are *rab-3p::tomm-20::mScarlet*. For panels (**B**–**K**), Neur-67Q refers to Neur-67Q worms expressing mitochondrially targeted mScarlet (Neur-67Q; *rab-3p::tomm-20::mScarlet* worms). Three biological replicates were performed. Statistical significance was assessed using a *t*-test. Error bars indicate SEM. * *p* < 0.05, *** *p* < 0.001.

**Figure 2 ijms-22-13447-f002:**
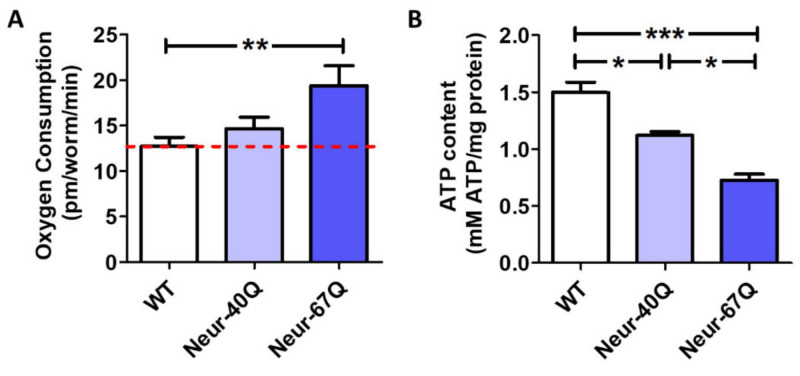
CAG repeat expansion in neurons disrupts mitochondrial function. Mitochondrial function in Neur-67Q worms was assessed by quantifying oxygen consumption and ATP levels in whole worms at day 1 of adulthood. Day 1 adult Neur-67Q worms have increased oxygen consumption (**A**) and decreased ATP levels (**B**) compared to wild-type worms. A minimum of three biological replicates were performed. Statistical significance was assessed using a one-way ANOVA with Bonferroni’s multiple comparison test. Error bars indicate SEM. * *p* < 0.05, ** *p* < 0.01, *** *p* < 0.001.

**Figure 3 ijms-22-13447-f003:**
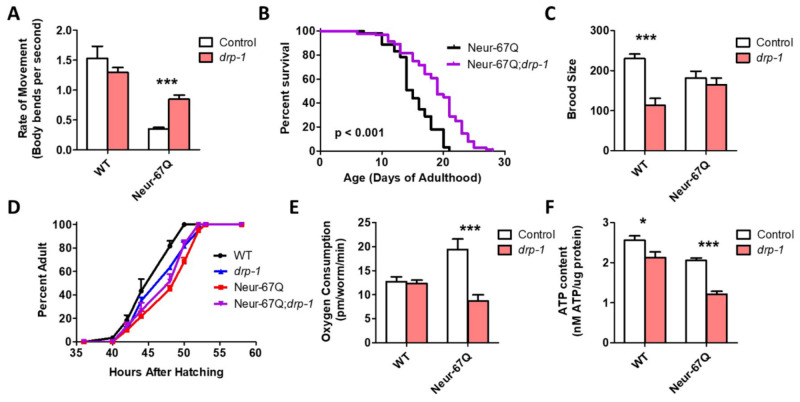
Inhibition of mitochondrial fission is beneficial in a neuronal model of polyglutamine toxicity. To examine the effect of disrupting mitochondrial fission in a neuronal model of polyglutamine toxicity, Neur-67Q worms were crossed to *drp-1* deletion mutants. Deletion of *drp-1* partially ameliorated phenotypic deficits in Neur-67Q worms. Neur-67Q;*drp-1* worms showed significantly increased movement (**A**) and lifespan (**B**) compared to Neur-67Q worms. Unlike wild-type worms, deletion of *drp-1* did not decrease fertility (**C**) or development time (**D**) in Neur-67Q worms. Combined, this indicates that inhibiting mitochondrial fission is beneficial in a neuronal model of polyglutamine toxicity. Neur-67Q worms have increased oxygen consumption compared to wild-type worms, and a mutation in *drp-1* decreases oxygen consumption in these worms (**E**). Deletion of *drp-1* causes a decrease in ATP levels in both wild-type and Neur-67Q worms (**F**). Control data for wild-type and *drp-1* worms was previously published in Machiela et al., 2021, as experiments for both papers were performed simultaneously using the same controls. For panels (**A**,**C**,**E**,**F**), “Control” refers to worms in a wild-type background with a normal expression of *drp-1* (either wild-type or Neur-67Q worms). A minimum of three biological replicates were performed. Statistical significance was assessed using a two-way ANOVA with Bonferroni posttest (panels (**A**,**C**,**E**,**F**)), the log-rank test (panel (**B**)), or a repeated measures ANOVA (panel (**D**)). Error bars indicate SEM. * *p* < 0.05, *** *p* < 0.001.

**Figure 4 ijms-22-13447-f004:**
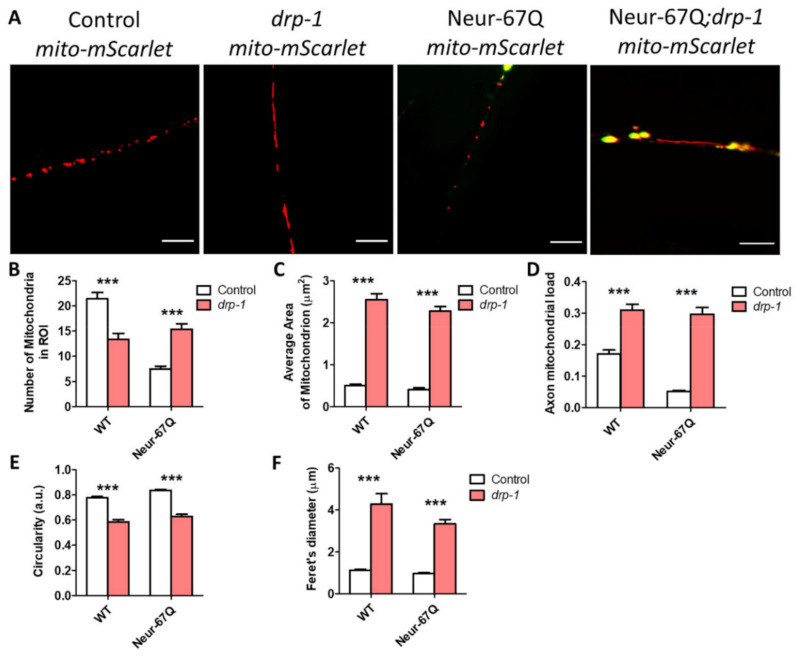
Disruption of *drp-1* rescues deficits in mitochondrial morphology caused by CAG repeat expansion. Deletion of *drp-1* decreased mitochondrial fragmentation in Neur-67Q and control worms at day 7 of adulthood. Representative images of Neur-67Q worms and control worms in wild-type and *drp-1* deletion background (**A**). Scale bar indicates 10 µM. Disruption of *drp-1* in Neur-67Q worms increased mitochondrial number (**B**), increased mitochondrial area (**C**), increased axonal mitochondrial load (**D**), decreased mitochondrial circularity (**E**), and increased the Feret’s diameter of the mitochondria (**F**). Control worms are *rab-3p::tomm-20::mScarlet*. For panels (**B**–**F**), “Control” refers to worms in a wild-type background with normal expression of *drp-1* (either wild-type or Neur-67Q worms). Control data is from Figure 1 and is shown to facilitate a direct comparison of the effects of *drp-1* deletion. Three biological replicates were performed. Statistical significance was assessed using a two-way ANOVA with Bonferroni post-test. Error bars indicate SEM. *** *p* < 0.001.

**Figure 5 ijms-22-13447-f005:**
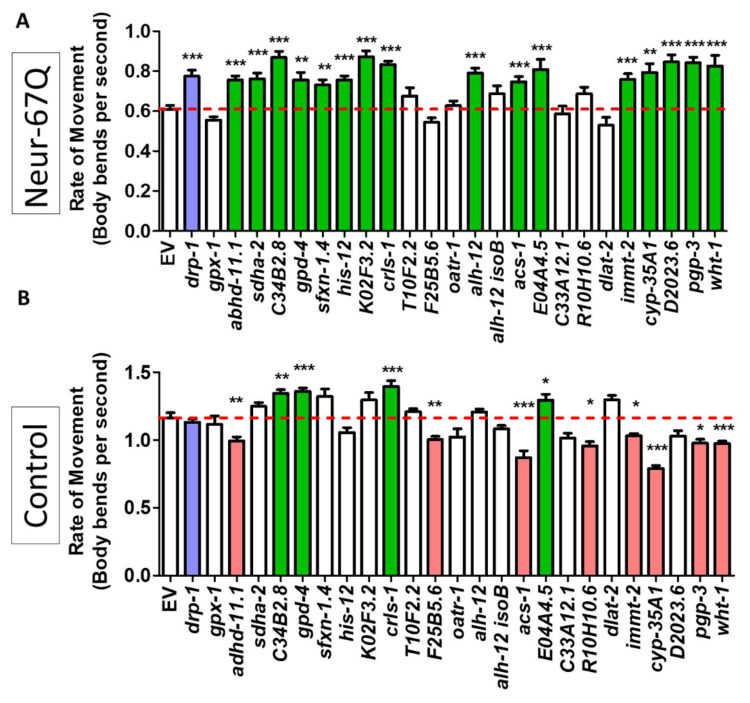
Decreasing mitochondrial fragmentation improves rate of movement in a neuronal model of polyglutamine toxicity. Neur-67Q worms in which RNAi is only effective in neurons (Neur-67Q*;sid-1;unc-119p::sid-1* worms) and a neuron-specific RNAi control strain (*sid-1;unc-119p::sid-1* worms) were treated with RNAi clones that decrease mitochondrial fragmentation in body wall muscle. RNAi against 16 of the 24 genes tested improved the rate of movement in Neur-67Q worms (**A**). RNAi against four of these genes also increased movement in the neuron specific RNAi strain (**B**). Green indicates a significant increase in movement, while red indicates a significant decrease in movement. The positive control *drp-1* is indicated with blue. “Neur-67Q” refers to Neur-67Q*;sid-1;unc-119p::sid-1* worms (panel (**A**)), while “Control” refers to *sid-1;unc-119p::sid-1* worms (panel (**B**)). Three biological replicates were performed. Statistical significance was assessed using a one-way ANOVA with Dunnett’s multiple comparison test. Error bars indicate SEM. * *p* < 0.05, ** *p* < 0.01, *** *p* < 0.001.

**Figure 6 ijms-22-13447-f006:**
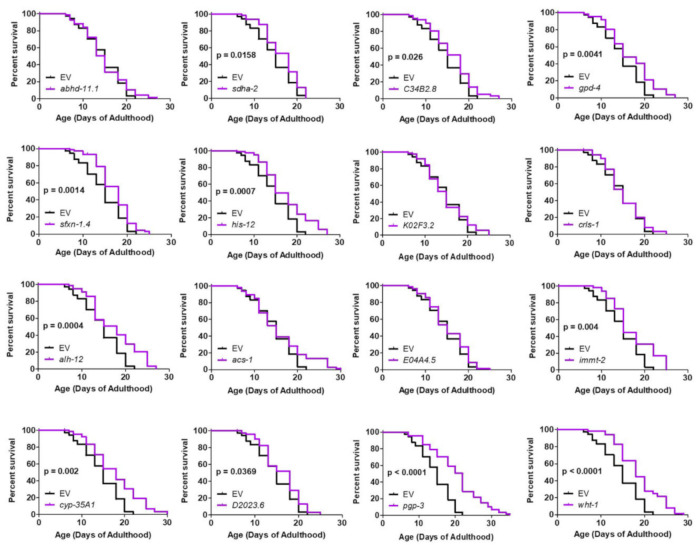
RNAi clones previously shown to decrease mitochondrial fragmentation in body wall muscle rescue shortened lifespan in neuronal model of polyglutamine toxicity. Neur-67Q worms in which RNAi is only effective in neurons (Neur-67Q*;sid-1;unc-119p::sid-1* worms) were treated with RNAi clones that decrease mitochondrial fragmentation in body wall muscle and that we found to increase movement in Neur-67Q worms (Figure 5). Eleven of the sixteen RNAi clones that improved movement in Neur-67Q worms also resulted in increased lifespan. Each of these RNAi clones targets a different gene. Three biological replicates were performed. Statistical significance was assessed using the log-rank test.

**Figure 7 ijms-22-13447-f007:**
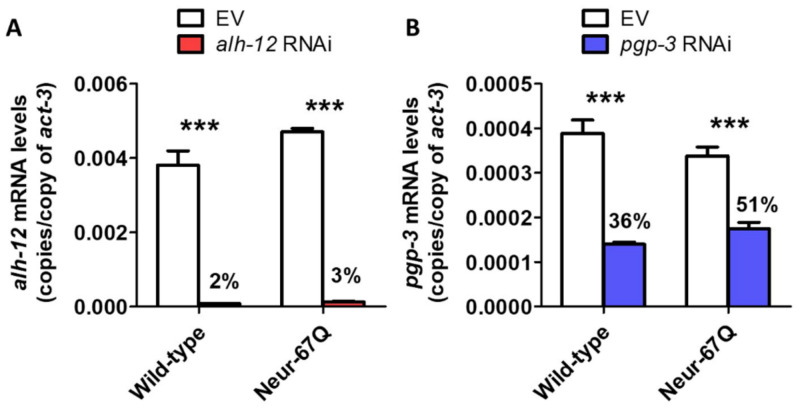
Decreasing the levels of *alh-12* and *pgp-3* mRNA using RNA interference. Wild-type and Neur-67Q worms were treated with RNAi bacteria targeting *alh-12* (**A**) or *pgp-3* (**B)** beginning at the L4 stage of the parental generation. mRNA levels were measured when the progeny reached the young adult stage using quantitative RT-PCR. In both wild-type and Neur-67Q worms, there was a significant decrease in *alh-12* and *pgp-3* mRNA levels when treated with RNAi bacteria targeting *alh-12* or *pgp-3*, respectively. The level of knockdown was highly significant for both genes, but the magnitude of knockdown was greater for *alh-12*. Bars indicate the mean value of three biological replicates. Statistical significance was assessed using a two-way ANOVA with Bonferroni post-test. Error bars indicate SEM. *** *p* < 0.001.

**Table 1 ijms-22-13447-t001:** Effect of RNAi clones that decrease mitochondrial fragmentation in neuronal and body wall muscle models of polyglutamine toxicity. “ND” indicates not done. “=” indicates no change. Data from BW-Htt74Q worms and BW-Htt28Q worms is from Machiela et al. 2021, *Aging and Disease*, PMID: 34631219).

Target Gene	*Drosophila* Homolog	Mammalian Homolog	Effect on Thrashing in Neur-67Q Worms	Effect on Lifespan in Neur-67Q Worms	Effect on Crawling in BW-Htt74Q Worms	Effect on Thrashing in BW-Htt74Q Worms	Effect on Thrashing in Neuron Specific RNAi Strain	Effect on Lifespan in Neuron Specific RNAi Strain	Effect of Crawling in BW-Htt28Q Worms	Effect of Thrashing in BW-Htt28Q Worms
*alh-12*	*Aldh*	*ALDH9A1*	**Increased**	**Increased**	**Increased**	**Increased**	No effect	**Decreased**	No effect	**Decreased**
*pgp-3*	*Mdr49*	*ABCB4*	**Increased**	**Increased**	**Increased**	**Increased**	**Decreased**	**Increased**	No effect	**Decreased**
*gpd-4*	*Gapdh2*	*GAPDH*	**Increased**	**Increased**	**Increased**	No effect	**Increased**	No effect	No effect	**Decreased**
*immt-2*	*Mitofilin*	*IMMT*	**Increased**	**Increased**	No effect	**Increased**	**Decreased**	**Increased**	No effect	No effect
*sdha-2*	*SdhA*	*SdhA*	**Increased**	**Increased**	**Increased**	No effect	No effect	**Increased**	**Increased**	**Decreased**
*wht-1*	*w*	*ABCG1*	**Increased**	**Increased**	**Increased**	No effect	**Decreased**	**Decreased**	No effect	No effect
*C34B2.8*	*ND-B16.6*	*NDUFA13*	**Increased**	**Increased**	**Increased**	**Decreased**	**Increased**	No effect	No effect	No effect
*drp-1*	*Drp1*	*DNM1L*	**Increased**	**Increased**	No effect	No effect	No effect	No effect	No effect	No effect
*F25B5.6*	*Fpgs*	*FPGS*	No effect	ND	**Increased**	**Increased**	**Decreased**	ND	No effect	No effect
*his-12*	*His2A*	*HIS2H2AB*	**Increased**	**Increased**	No effect	No effect	=	**Decreased**	**Decreased**	**Decreased**
*sfxn-1.4*	*Sfxn1-3*	*SFXN1/3*	**Increased**	**Increased**	No effect	No effect	=	**Decreased**	No effect	No effect
*abhd-11.1*	*CG2059*	*ABHD11*	**Increased**	=	No effect	No effect	**Decreased**	=	No effect	No effect
*acs-1*	*Acsf2*	*ACSF2*	**Increased**	=	No effect	No effect	**Decreased**	=	**Decreased**	No effect
*crls-1*	*CLS*	*CRLS1*	**Increased**	=	No effect	No effect	**Increased**	=	No effect	No effect
*cyp-35A1*	*Cyp18a1*	*CYP2C8*	**Increased**	**Increased**	**Decreased**	No effect	**Decreased**	=	No effect	No effect
*D2023.6*	*Adck1*	*ADCK1*	**Increased**	**Increased**	**Decreased**	No effect	=	=	**Decreased**	No effect
*dlat-2*	*muc*	*DLAT*	No effect	ND	**Increased**	No effect	No effect	ND	**Decreased**	**Decreased**
*gpx-1*	*PHGPx*	*GPX4*	No effect	ND	No effect	**Increased**	No effect	ND	No effect	No effect
*timm-17B.1*	*Tim17b*	*TIMM17A/B*	**Increased**	=	No effect	No effect	**Increased**	=	No effect	**Decreased**
*oatr-1*	*Oat*	*OAT*	No effect	ND	**Increased**	No effect	No effect	ND	**Increased**	No effect
*R10H10.6*	*CG2846*	*RFK*	No effect	ND	**Increased**	No effect	**Decreased**	ND	No effect	No effect
*alh-12 iso B*	*Aldh*	*ALDH9A1*	=	ND	**Decreased**	No effect	=	ND	No effect	No effect
*C33A12.1*	*ND-13B*	*NDUFA5*	=	ND	No effect	No effect	=	ND	No effect	No effect
*K02F3.2*	*aralar1*	*SLC25A12*	**Increased**	=	**Decreased**	No effect	=	=	No effect	**Decreased**
*T10F2.2*	*CG1628*	*SLC25A15*	=	ND	No effect	No effect	=	ND	No effect	No effect

## Data Availability

All data is available upon request.

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
