# Peer review of "Identification of Novel Therapeutic Targets for Polyglutamine Diseases That Target Mitochondrial Fragmentation"

_ijms, 2021, doi:10.3390/ijms222413447_

Round 1
Reviewer 1 Report
In this study, Traa and colleagues propose to identify several siRNA molecules for their ability to decrease mitochondrial fragmentation, as a possible therapeutic strategy to treat polyglutamine disorders toxicity. The study is interesting and relevant for the field of polyglutamine diseases, in which mitochondrial dysfunction seems to be important. However, there are several issues that prevent the publication of the manuscript in the current form and for that, I advise the authors to address them and reformulate the manuscript.
Major issues
Figure 1A and S1A are the same, although in the text they are always mentioned as supplementary. Probably it was a typo, but it must be corrected. It is already strange that the graphs while representing the same, they have different scales and apparently different quantifications.
Figure 1 – what is labelled in green? Figure C is significantly different, however, in the text, it says that no difference was found (lines 99-100). It would be helpful to show neuronal marker labeling also.
Figure S2 is not what is mentioned in the text. Overall, there is a mislabeling of the images, which makes the paper quite hard to read.
Line 148-149 – the phrase is quite speculative, and the results shown do not support entirely what is stated.
Line 224-225 – the phrase is quite speculative, and the results shown do not support entirely what is stated.
Table S1 – it is not clear the rationale of this table and how the results of the BW-Htt74Q model were obtained (this last part is also applicable to table 1)
Line 329-344) – while it is important to discuss the importance of top-ranked target genes, a concluding remark for the discussion is mandatory, otherwise, these lines are more in line with an introductory aspect than with a discussion.
Minor issues
Line 50 – references 5,6 are about HD and therefore for generalizing the phrase to polyglutamine disorders other studies should be mentioned or at least a review focusing on this aspect in different polyglutamine disorders.
Figure 3 – the size bar of the images is missing
Material and methods – The RNAi clones’ origin is not mentioned
Reviewer 2 Report
The study by Traa et al uses a worm model of polyQ disease to investigate the effect of decreasing mitochondrial fragmentation, using deletion of the fission protein DRP-1 and also 24 RNAi clones. The main ideas are that excessive mitochondrial fission is deleterious in polyQ disease, and that its inhibition by deleting/disrupting DRP-1 was previously found protective in several models, but can be detrimental as shown by the authors in a worm model. Thus, the authors are looking for other non-DRP-1 targets to reduce mitochondrial fragmentation, as a therapeutic strategy for polyQ disease. The concept is interesting, and the authors have experience in the field, namely with worm models of neurodegeneration. There are however issues with the controls, validation of the targets, and the mechanistic explanation for the findings. As it stands, the study is mainly observational and preliminary, since the main novel message (identification of new targets) still requires validation. Below are suggestions to improve the study:
- Update the supporting literature. Most of the cited references have over 5 years, and this is particularly true for previous studies supporting the interest of targeting DRP-1, and the library of RNAi used.
- Reconsider the positioning of supplementary data - some are key to characterize the model phenotypes; and consider including representative direct experimental data (e.g. respirometry tracings from the Seahorse) to complement bar graphs (indirect data).
- Improve the discussion of results, particularly where there are contradictory/unexpected findings such as: decreased total ATP with DRP-1 deletion being associated with increased motility and survival; DRP-1 deletion reducing fertility by 50% in control but having no effect on that of Neur-67Q; alh-12 and pgp-13 RNAi having different effects across C. elegans strains.
- Clarify what exactly are the controls (sometimes mentioned “control”, others “wild-type”), particularly when the Neuro-67Q has been crossed with other strains (with fluorescent mitochondria, or increased RNAi efficiency). This is particularly important since the same treatment in different strains can either increase or decrease mobility or survival (e.g. compare alh-12 and pgp-13 RNAi effects across the strains in Table 1).
- Validate the key positive RNAi hits by qPCR confirmation (ideally also by Western Blot, and by showing it can be rescued). The authors justify in page 11, lines 307-308, not doing this validation due to a high number of positive hits. However, the qPCR control should have been done at least for the 2 most promising targets identified by the authors alh-12 and pgp-13 (page 11, lines 323-324).
- Identify or at least propose literature supported and mechanistic hypothesis for the new targets. As it stands, for the main targets (alh-12 and pgp13), it is simply stated that “it is hard to speculate how disrupting alh-12 may be acting to improve movement and lifespan in the worm models of polyQ toxicity (page 12 lines 332-333)”. There is no discussion to the finding in Table 1 that alh-12 RNAi decreased life span and movement in other strains. Also, for pgp-13, the authors simply state it is unclear how disrupting a protective target would protect against polyQ, and speculate it could involve hormesis (page 12, lines 341-342). Without qPCR validation, and without mechanistic hypothesis, the main message of the paper – identification of new targets – is difficult to support.
Reviewer 3 Report
Manuscript by Traa et al. provides experimental insights into selection of potential therapeutic targets which could diminish neurological effects connected with mitochondrial dysfunction in Huntington’s disease (HD). The authors used C. elegans model, established 15 years ago, to confirm that disruption of drp-1 is not a feasible therapeutic option, and screen additional 24 clones, with other genes disrupted, to find the ones which silencing could provide beneficial therapeutic effects in HD without prominent adverse effects.
Major comments:
- The results could have more clear interpretation and could provide more convincing data if the control model also contained CAG repeat tract in the transgene, for example 19Q – the one of control models in the original study describing C. elegans model used.
- Validation of identified therapeutic targets in other model would provide much greater impact of this study and would prove the significance of the findings.
Minor comments:
- “polyQ” is a commonly used abbreviation for “polyglutamine”, it should be introduced at the beginning and used throughout the text
- also using a name: “polyglutamine diseases”, instead of less commonly used: “polyglutamine toxicity disorders” should be considered
- Abstract, lines: 24-25, it should be more clearly stated that eleven genes were identified, instead of clones. It is not clear here that each clone represents other RNAi-based disruption of gene
- Lines 47-75 and 87: It is stated that model used is “well-characterized” – it should be more clearly indicated that the model have not been characterized before for mitochondrial features
Round 2
Reviewer 1 Report
Thank you for addressing my comments.
Author Response
Dear Reviewer 1,
Thank you for the positive review of our manuscript.
Sincerely,
Jeremy Van Raamsdonk
Reviewer 2 Report
Although the revised manuscript has improved, there are two main issues that should be addressed before the manuscript is suitable for publication.
The authors should be given more time (e.g. 1-2 months, if necessary) to complete and include the following results in the main Figures:
1) Validation of the two key RNAi hits (pgp-3 and alh-12) with qPCR (the figure proposed as S7 should be a main Figure) in order to provide experimental evidence that supports the main conclusions.
2) As the authors mention they "no longer have access to the respirometry tracings", they should repeat at least one key experiment with the Seahorse to provide direct experimental data (respirometry tracings) to support the main statements about effects on the "rate of oxidative phosphorylation (oxygen consumption)" (Section 2.3, line 154).
Reviewer 3 Report
Manuscript has been largely improved. For further research plans I suggest to consider very deeply what controls should be used. Results and findings which are supported with proper controls will always have greater importance and impact.
Author Response
Dear Reviewer 3,
Thank you for the positive review of our manuscript and your advice regarding the careful consideration of controls.
Sincerely,
Jeremy Van Raamsdonk